# Prolonged Extracorporeal Circulation Leads to Inflammation and Higher Expression of Mediators of Vascular Permeability Through Activation of STAT3 Signaling Pathway in Macrophages

**DOI:** 10.3390/ijms252212398

**Published:** 2024-11-19

**Authors:** Jana Luecht, Camila Pauli, Raphael Seiler, Alexa-Leona Herre, Liliya Brankova, Felix Berger, Katharina R. L. Schmitt, Giang Tong

**Affiliations:** 1Department of Congenital Heart Disease/Pediatric Cardiology, Deutsches Herzzentrum der Charité, 13353 Berlin, Germany; jana.luecht@dhzc-charite.de (J.L.); rseiler@dhzc-charite.de (R.S.); felix.berger@dhzc-charite.de (F.B.); 2Department of Developmental Pediatric Cardiology, Charité—Universitätsmedizin Berlin, 10117 Berlin, Germany; camila.pauli@charite.de (C.P.); alexa-leona.herre@charite.de (A.-L.H.); lyliya.brankova@charite.de (L.B.); 3Department of Developmental Pediatric Cardiology, Deutsches Herzzentrum der Charité, 13353 Berlin, Germany; katharina.schmitt@dhzc-charite.de

**Keywords:** congenital heart defects, cardiopulmonary bypass, macrophages, sterile inflammation, vascular permeability, STAT3, NF-κB p65, Stattic, DAMPs, cytokines

## Abstract

Congenital heart defects (CHDs) are one of the most common congenital malformations and often require heart surgery with cardiopulmonary bypass (CPB). Children undergoing cardiac surgery with CPB are especially at greater risk of post-operative complications due to a systemic inflammatory response caused by innate inflammatory mediators. However, the pathophysiological response is not fully understood and warrants further investigation. Therefore, we investigated the inflammatory response in macrophages initiated by peri-operative serum samples obtained from patients with CHD undergoing CPB cardiac surgery. Human differentiated THP-1 macrophages were pretreated with Stattic, a STAT3 (Tyr705) inhibitor, before stimulation with serum samples. STAT3 and NF-κB activation were investigated via a Western blot, IL-1β, TNFα, IL-10, mediators for vascular permeability (VEGF-A, ICAM), and SOCS3 gene expressions via RT-qPCR. CPB induced an inflammatory response in macrophages via the activation of the STAT3 but not NF-κB signaling pathway. Longer duration on the CPB correlated with increased cytokine, VEGF, and ICAM expressions, relative to individual pre-operation levels. Patients that did not require CPB showed no significant immune response. Pretreatment with Stattic significantly attenuated all inflammatory mediators investigated except for TNFα in the macrophages. CPB induces an increased expression of cytokines and mediators of vascular permeability via the activation of STAT3 by IL-6 and IL-8 in the serum samples. Stattic attenuates all mediators investigated but promotes TNFα expression.

## 1. Introduction

Congenital heart defects (CHDs) are one of the most common congenital malformations and often require heart surgery with cardiopulmonary bypass (CPB) [1]. Among others, the operation itself and CPB-induced ischemia/reperfusion injury lead to the release of inflammatory mediators and damage-associated patterns (DAMPs) into the bloodstream, activating various cascades (coagulation cascade, complement system, immune system) and inducing an acute inflammatory response, which may lead to post-operative complications such as systemic inflammatory response syndrome (SIRS) [2,3] and capillary leak syndrome (CLS) [4,5]. Despite technological progress, the harmful influences of CPB in neonates and children are often pronounced due to their immature tissue, organ function, and immune system. Moreover, the disparity in size between the CPB circuit and patients’ cardiovascular system surface increases the risk for complications [6,7,8], resulting in higher morbidity with prolonged stays in the intensive care unit (ICU), prolonged ventilation times and sedation, increased drainage losses, and an increased need for diuretics and catecholamines [9,10]. Furthermore, the group of surviving adults with congenital heart defects (ACHDs) is growing thanks to medical advances and treatments. ACHDs represent an unique patient group with distinct physiological characteristics that may impact both their surgical outcomes and inflammatory responses to CPB [11,12].

To minimize complications in heart surgery, a multimodal approach is necessary due to the complex pathogenesis [13]. There have been further developments in various areas, including neuroprotective strategies, improved kidney and blood coagulation management, and measures in infection prevention specifically aimed at reducing the inflammatory response [14]. CPB systems optimized for children with shorter tubes of a smaller diameter, coating of the tubes with biocompatible substances such as heparin, and further development of surgical techniques that enable shorter operating times have led to a significant improvement in the surgical treatment of congenital heart defects and safer pediatric cardiac surgery with lower complication rates [13,15,16,17].

However, measures that intervene directly with the immune response such as the pre-operative administration of glucocorticoids, which is established in many centers, are viewed critically. There are currently no standardized guidelines. The optimal dosage, time of administration, and specific patient groups that may profit from glucocorticoids and many other factors remain unclear [18,19,20,21,22]. Moreover, there is a general lack of data from both pediatric and ACHD patients since previous studies have mainly focused on adults with non-congenital heart defects.

Regarding cellular signaling pathways, the emphasis of many former studies has been on the transcription factor NF-κB [3,23,24]. Furthermore, the pathophysiology of the triggered inflammatory reaction and the associated complications is still not fully understood. In our study, we focus on the evolutionarily conserved JAK-STAT3 pathway, which can be activated by inflammatory mediators such as IL-6 [25]. The JAK-STAT3 pathway plays a major role in regulating the immune response by modulating the expression of various cytokines and mediators for vascular permeability [26,27]. Moreover, multiple crosstalk mechanisms are being described between the STAT3- and NFκB-signaling pathway. Lee et al. demonstrate a constitutive NFκB activity maintained by STAT3 in cancer cells via complex formation with NFκB subunits and the direct modulation of the gene expression of components important for NFκB signaling [28]. IL-6 is released by different cell types including macrophages, T cells, and endothelial cells, and is an important mediator in acute inflammatory reactions by modulating the balance between pro- and anti-inflammatory mediators; by interfering in the cell growth, survival, and differentiation; or by promoting the production of acute-phase proteins [29,30]. Particularly in pediatric patients, IL-6 is already an established biomarker for Sepsis [31,32].

The goal of our exploratory study is to better understand the heterogenous patient population who underwent corrective or palliative heart surgery for CHD at our hospital, as well as to shed light on the complex pathophysiology of CPB-induced inflammation in this cohort. Therefore, we optimized THP-1 human macrophages differentiated from acute monocytic leukemia cells by the PMA model to investigate the in vitro inflammatory response induced by the secreted mediators in peri-operatively collected serum samples from our patient cohort with CHD who underwent cardiac surgery with and without CPB. Moreover, we aim to investigate if there is a correlation between CPB duration and inflammatory response with a focus on the activation of the JAK-STAT3 pathway.

## 2. Results

In the current study, we enrolled 56 consecutive patients whose demographic and clinical data are summarized in Table 1. The study cohort consisted of 31 males of a median age of 2.3 years (range: 0 to 59 years) and 25 females of a median age of 3.6 years (range: 0 to 65 years). Our heterogeneous cohort included patients with a broad range of ages (47 patients between 0 and 18 years old and 9 patients between 18 and 65 years old), congenital heart diseases, and a complexity of anomalies treated in our hospital. In order to collect a representative cohort for patients with CHD, we included patients receiving their first and corrective operation such as closure of simple atrial or ventricular septal defects, as well as patients undergoing palliative surgery such as Norwood I, pulmonal atrial banding, or Fontan procedures. No neonatal patients were included in the study and all patients survived until hospital discharge.

To quantify the complexity of the surgical procedure based on risk adjustment concerning in-hospital mortality, we used Risk Adjustment for Congenital Heart Surgery 1 (RACHS-1) [33]. RACHS-1 could not be assessed for one 2-year-old patient as the cardiac surgery was performed to implant a two-chamber pacemaker in the patient with a congenitally corrected transposition of the great arteries and third-degree congenital atrioventricular block, and this kind of operation is not included in this risk score.

Severity of CHD was classified according to the three categories suggested by Warnes et al. [12]: complex (e.g., the transposition of the great arteries, pulmonary atresia, tricuspid atresia, hypoplastic left heart syndrome), moderate (e.g., atrioventricular canal defects, tetralogy of Fallot, the coarctation of the aorta, ventricular septal defects with associated lesions), and mild (e.g., isolated small atrial septal defects or isolated small ventricular septal defect without associated lesions, isolated congenital aortic valve disease).

### 2.1. Schedule for Peri-Operative Serum Sample Collection

Serum samples from our patient cohort were collected as previously described [34]. Briefly, blood samples were obtained via the central venous line pre-operatively after the induction of anesthesia (T0), post-operatively upon arrival in the pediatric intensive care unit (PICU) (T1), 6 h after the operation (T2), and 24 h after the operation (T3), as illustrated in Figure 1.

### 2.2. CPB Induces Activation of STAT3 but Not NF-κB p65 Signaling Pathway in THP-1 Macrophages In Vitro

As previously reported, we detected significantly elevated levels of IL-6, IL-8, and IL-10 concentrations in the serum samples from patients with CHD undergoing cardiac surgery with CPB at all post-operative timepoints investigated (T1–T3), relative to individual baseline values (T0) [34]. Since both IL-6 and IL-8 are known to phosphorylate the STAT3 and IL-10 is known to inhibit the NF-κB p65 signaling pathways, we investigated the inflammatory signaling pathways induced in the THP-1 macrophages by these inflammatory mediators in our CHD cohort. We observed a significant increase in STAT3 phosphorylation at the Tyr705 site post-operatively, relative to individual baseline values, as shown in Figure 2a. Post-operative serum-mediated STAT3 phosphorylation was significantly increased immediately after surgery upon arrival in the pediatric intensive care unit (PICU) (T1, *p* = 0.0363), then gradually returned to individual baseline values over the 24 h observation period after the operation. No significant increase in STAT3 phosphorylation at the Ser727 site was observed. Moreover, pretreatment with 5 µM Stattic significantly attenuated STAT3 activation at T1 (*p* = 0.0248) in the macrophages. Interestingly, we did not observe any significant increase in NF-κB p65 phosphorylation induced by post-operative CPB serum samples, as shown in Figure 2b. In fact, post-operative values were, in general, lower than individual baseline values, and pretreatment with Stattic significantly reduced NF-κB p65 phosphorylation 6 h after the operation (T2, *p* = 0.0162). Moreover, we also observed a significant attenuation of NF-κB p65 phosphorylation by Stattic in our LPS-stimulated experimental positive control (*p* = 0.0038).

### 2.3. Prolonged Duration of CPB Correlated with Induced Inflammatory Response and Increased Expression of Mediators of Vascular Permeability in THP-1 Macrophages In Vitro

We observed that the peri-operative kinetics of the expression of cytokines and mediators of vascular permeability induced by patients’ serum in THP-1 macrophages in vitro can be correlated with the duration of CPB, as shown in Figure 3. The patient cohort was classified into four distinct groups based on the duration of CPB as follows: 0 h (*n* = 5), 1–3 h (*n* = 32), 4–6 h (*n* = 13), and 7–9 h (*n* = 6). Patients undergoing cardiac surgery without CPB (0 h) did not elicit a significant immune response post-operatively in the macrophages for all targets investigated. Increased IL-1β expression in the macrophages correlated with an increased duration of CPB immediately after operation upon arrival in the pediatric intensive care unit (PICU) that was significantly higher for the 7–9 h CPB group (T1, *p* = 0.0283) relative to patients operated on without CPB (0 h), as shown in Figure 3a. A moderate duration of CPB, 1–3 h and 4–6 h, elicited a continuous increase in IL-1β expression throughout the post-operative phase that ultimately peaked at 24 h in both groups but did not reach significancy. Similarly, we observed that induced TNFα expression in the macrophages also continuously increased throughout the post-operative phase and peaked at 24 h in both 1–3 h and 4–6 h CPB groups but did not reach significancy relative to patients without CPB, as shown in Figure 3b. Interestingly, TNFα expression in the 7–9 h CPB group did not increase and was similar to the 0 h CPB group. Induced IL-10 expression for the 7–9 h CPB group was significantly higher than patients operated on without CPB immediately after the operation at T1 (*p* = 0.0291) and remained significantly higher 6 h after the operation (T2, *p* = 0.0284) before returning to baseline, as shown in Figure 3c. IL-10 expression for the 4–6 h CPB group continuously increased throughout the post-operative phase and reached significancy at both 6 h (T2, *p* = 0.0243) and 24 h (T3, *p* = 0.0254).

The elicit expression of the mediators of vascular permeability, including intracellular adhesion molecule-1 (ICAM) and vascular endothelial growth factor (VEGF), as well as the suppressor of cytokine signaling 3 (SOCS3), in the macrophages also showed similar correlation with the duration of CPB. We observed an immediate significant spike in SOCS3 expression relative to patients without CPB at T1 (*p* = 0.0015) that decreased towards baseline but remained significant 6 h after the operation (T2, *p* = 0.0375), as shown in Figure 3d. Interestingly, we did not observe any induced ICAM expression in the macrophages in our entire cohort, as shown in Figure 3e. ICAM expression in the 4–6 h CPB group continuously increased throughout the post-operative phase, which peaked at 24 h but did not reach significancy. Induced VEGF expression kinetics in the macrophages were similar to that of SOCS3, with a significant spike in the 7–9 h CPB group immediately after the operation at T1 (*p* = 0.0005) that decreased back to baseline but remained significant 6 h after the operation (T2, *p* = 0.0009), as shown in Figure 3f.

In summary, increasing the duration of CPB from 1 to 6 h elicited an immune response that gradually increased post-operatively upon reaching significancy relative to individual baseline values 24 h after the operation. In general, the 4–6 h CPB group showed a higher immune response than the shorter 1–3 h group. On the other hand, an extreme duration of CPB lasting from 7 to 9 h resulted in an immediate induced immune response in the macrophages upon arrival at the ICU (T1) that gradually decreased towards baseline.

### 2.4. Stattic Attenuates CPB-Induced Cytokines and Mediators of Vascular Permiability but Promotes TNFα Expression in THP-1 Macrophages In Vitro

Cardiac surgery requiring the assistance of CPB to correct congenital heart defects is known to induce a systemic inflammatory response in patients. Based on our experimental observations and previous reports of the role of STAT3 in post-operative systemic inflammation [35], we investigated the role of CPB in the induction of the expression of cytokines and mediators of vascular permeability, as well as the effect of pretreatment with 5 µM Stattic on this inflammatory response, in an in vitro model of human THP-1 macrophages. Serum samples collected from patients were categorized according to the duration of CPB: no CPB (0 h) and 1–3 h, 4–6 h, and 7–9 h CPB. In general, we observed that serum samples from patients undergoing heart operation with the assistance of CPB elicited higher cytokine expressions than patients that did not require CPB, as shown in Figure 4. Additionally, the data are also shown in sub-groups with respect to patients’ age, children (0–18 yrs old) and adults (18–65 yrs old), in Appendix A. Interestingly, serum-induced IL-1β and IL-10 expressions were significantly lower upon arrival at the ICU after the operation (T1) then individual baseline values before the operation (T0) in patients not requiring CPB, as shown in Figure 4a and Figure 4c, respectively. We observed a significant decrease in IL-1β expression in patients not requiring CPB (0 h) at T1 (*p* = 0.0260), as shown in Figure 4a. However, for patients undergoing CPB, we observed a significant increase in IL-1β expression in the 1–3 h CPB group at T3 (*p* = 0.0268) and in the 4–6 h group at T3 (*p* = 0.0216). Stattic significantly attenuated IL-1β expression in the 4–6 h CPB group at T0 (*p* = 0.0231), T1 (*p* = 0.0197), and T3 (*p* = 0.0158).

We also observed a significant increase in TNFα expression in the 1–3 h CPB group at T3 (*p* = 0.0097) but a significant decrease in the 1–9 h CPB group at T1 (*p* = 0.0273), as shown in Figure 4b. Interestingly, Stattic significantly augmented TNFα expression in the 1–3 h CPB group at all time points investigated, T0 (*p* < 0.0001), T1 (*p* = 0.0029), T2 (*p* < 0.0001), and T3 (*p* = 0.0002); in the 4–6 h CPB group at T0 (*p* = 0.0015), T1 (*p* = 0.0332), and T3 (*p* = 0.0220); and in the 7–9 h CPB group only at T1 (*p* = 0.0237).

IL-10 expression showed similar expression kinetics to IL-1β expression. We observed a significant decrease in IL-10 expression in patients not requiring CPB (0 h) at T1 (*p* = 0.0128), as shown in Figure 4c. Moreover, for patients undergoing CPB, we observed a significant increase in IL-10 expression in the 1–3 h CPB group at T3 (*p* = 0.0299) and in the 4–6 h group at T2 (*p* = 0.0193) and T3 (*p* = 0.0086). Stattic significantly attenuated IL-10 expression in the 0 h CPB group at T0 (*p* = 0.0049), in the 1–3 h CPB group at T1 (*p* = 0.0058), in the 4–6 h CPB group at T3 (*p* = 0.0173), and in the 7–9 h CPB group at T2 (*p* = 0.0030).

We further investigate the expression of the mediators of vascular permeability, ICAM and VEGF, as well as the inhibitor of the Stat3 activation molecule, SOCS3. We observed a significant decrease in ICAM expression in patients not requiring CPB (0 h) at T1 (*p* = 0.0051), but no significant changes in patients undergoing CPB, as shown in Figure 5a. Interestingly, Stattic significantly increased ICAM expression in the 1–3 h CPB group at T0 (*p* = 0.0430) and T2 (*p* = 0.0207) but decreased ICAM expression in the 4–6 h CPB group at T1 (*p* = 0.0368). For VEGF expression, we observed a significant increase only in the 1–3 h CPB group at T3 (*p* = 0.0252), as shown in Figure 5b. Stattic significantly attenuated VEGF expression in the 1–3 h CPB group at T1 (*p* = 0.0004) and in the 4–6 h CPB group at T1 (*p* = 0.0057) and at T3 (*p* = 0.0091). SOCS3 expression was significantly upregulated in the 1–3 h CPB group at T1 (*p* = 0.0465), as shown in Figure 5c. Stattic increased SOCS3 in the 1–3 h CPB group at T1 (*p* = 0.0252). Additionally, the data are also shown in sub-groups with respect to patients’ age, children (0–18 yrs old) and adults (18–65 yrs old), in Appendix A.

## 3. Discussion

As previously reported, we detected significantly elevated levels of serum IL-6, IL-8, and IL-10 at all post-operative timepoints investigated (T1–T3) in patients with congenital heart defects undergoing cardiac surgery involving CPB [34]. Other clinical studies have also reported similar significant elevations in plasma cytokine levels in patients undergoing CPB [36,37]. Moreover, CPB-induced release of DAMPs such as mitochondrial DNA (mtDNA), which can also elicit a systemic immune response via toll-like receptor-9 signaling, has also been reported [38]. In our effort to investigate the underlying molecular mechanisms of inflammation in our in vitro model of CPB-induced macrophage activation, the significant increase in Stat3 and lack of NF-κB p65 activation indicate cytokines and not DAMPs as the dominant mediators of inflammation after the operation in our cohort. In fact, we observed that NF-κB p65 phosphorylation was lower immediately after the operation (T1) and continued to decrease relative to individual baseline levels before CPB (T0) (Figure 2a and Figure 2b, respectively).

The combined effect on the signaling pathways resulted in a significant increase in IL-1β, TNFα, and IL-10 expressions 24 h post-operatively (T3) for patients who underwent moderate durations of CPB. Blocking Stat3 activation by Stattic partially attenuated IL-1β and IL-10 expressions at some observed time points but significantly augmented TNFα expression at basically all investigated time points after the operation after stimulation with serum samples from the same patients (Figure 3a, Figure 3b, and Figure 3c, respectively). Our findings correlate with those of de Jong et al., who reported that the abrogation of the IL-10/STAT3 pathway restored in vitro in LPS-induced TNF-α production in human PBMC isolated from a pediatric cohort undergoing CPB-assisted surgery to correct simple congenital heart defects was independent of p38 MAPK attenuation or IκB-α degradation [35].

During the acute phase reaction following CPB, surgical trauma, ischemia–reperfusion injury, contact with the surface of the external circulatory system, and other triggers can induce the secretion of various pro- and anti-inflammatory cytokines and adhesion molecules, which is mediated by NF-kB signaling. Justifiably so, various therapeutic strategies emphasize the attenuation of NF-kB activation to minimize the systemic inflammatory response. However, Stat3 has also been shown to play a crucial role in the anti-inflammatory pathway [39,40]. Specifically, the anti-inflammatory properties of IL-10 and its effect on macrophage deactivation have been reported [38]. However, the direct effect of IL-10 on NF-κB activity is controversial. Murray proposed that IL-10 can indirectly inhibit NF-κB-driven gene transcription via the activation of Stat3 to reduce the overall transcriptional rate of specific genes [41]. Moreover, Stat3-deficient macrophages have been reported to have high sensitivity to a lipopolysaccharide, a potent mediator of NF-κB-driven inflammation. The authors also further concluded that the IL-10 activation of Stat3 is critically involved in the deactivation of macrophages and neutrophils [39].

Both VEGF and ICAM-1 not only play an important role in promoting tumor angiogenesis but they are also potent inducers of vascular permeability, which play a major role in the induction of endothelial dysfunction and the associated capillary leak syndrome. It is well known that VEGF activates the STAT3 signaling pathway, inducing a positive feedback loop [42,43,44]. Furthermore, it has been shown that IL-6 promotes the expression of VEGF, and thereby the inhibition of the IL6R-STAT3 signaling pathway leads to a reduced production of VEGF [45,46]. Regarding ICAM, STAT3 is known to promote its expression by binding to the ICAM promoter after being activated by inflammatory mediators such as IL-6 or TNFα [42,47].

Our findings suggest that STAT3 signaling plays a vital role in the systemic inflammatory response commonly observed after cardiac surgery requiring CPB. Moreover, the STAT3 modulation of ICAM-1 and VEGF is crucial in determining not only the inflammatory response, but also vascular health and tissue repair mechanisms during and after CPB. Similarly to our previous findings, we continue to observe large inter-individual differences in the inflammatory response in our patients’ cohort for all targets investigated [48]. Furthermore, we analyzed the influence of age in our previously published feasibility study including 19 patients ranging from 0 to 18 years of age [48], as well as in our follow-up study with 105 patients ranging from 0 to 65 years of age (data currently under preparation for publication). In both studies, we observed that patients’ age and sex had no significant influence on the investigated biomarkers for post-operative inflammation. Minimizing the post-operative sterile inflammatory response induced by cardiopulmonary bypass may significantly reduce the patient’s risk and improve the outcome. A multimodal approach is most likely needed for any therapeutic strategy and a focus on post-operative cytokine-induced STAT3 signaling should be included.

## 4. Materials and Methods

### 4.1. Protocol for Serum Sample Collection

The study was approved by the Ethics Committee of Charité—Universitätsmedizin Berlin, Germany (decision EA2/180/19). Serum samples from our patient cohort were collected as previously described [34]. Briefly, blood samples were obtained via the central venous line pre-operatively after the induction of anesthesia (T0), post-operatively upon arrival in the pediatric intensive care unit (PICU) (T1), and both 6 h (T2) and 24 h (T3) after the operation (Figure 1). We collected 1 mL blood from patients ≤15 kg and 2 mL of blood from patients >15 kg at each time point in Serum-Gel Microvette^®^ 500 (20.1344 Sarstedt, Nümbrecht, Germany). Samples were centrifuged at 10,000× *g* for 10 min and temporarily stored at −8 °C until transfer to final storage at −80 °C.

### 4.2. Cell Culture

THP-1 cells represent a human monocytic cell line that can be transformed to macrophages by stimulating with PMA, as described further below. By optimizing a standard transformation and activation protocol in our laboratory, the use of THP-1 cells allows us to obtain reproducible results that mimic human inflammatory responses under controlled conditions [49,50]. Immortalized human THP-1 cells were purchased from ATCC (American Type Culture Collection: TIB-202) and grown in suspension in an RPMI 1640 (Gibco, Thermo Fisher, Karlsruhe, Germany) medium supplemented with 10% (*v*/*v*) heat-inactivated fetal bovine serum (FBS, Merck Millipor, Billerica, USA) and 100 U/mL penicillin and 100 µg/mL streptomycin (Gibco, Thermo Fisher, Karlsruhe, Germany) in a humidified, 37 °C, 5% CO_2_ incubator. A total of 1 × 10^6^ cells were plated in 35 × 10 mm tissue culture-treated plates (Nunclon^TM^ Delta Surface, Thermo Scientific, Karlsruhe, Germany) in the presence of 15 ng/mL phorbol 12-myristiate-12 acetate (PMA, Sigma-Aldrich, Munich, Germany) for 3 days, followed by incubation in a PMA-free medium for 24 h prior to stimulation with serum samples or LPS. Differentiated THP-1 macrophages were then pretreated with 5 µM Stattic, a STAT3 (Tyr705) inhibitor, for 1 h, followed by incubation with serum samples, which were diluted with an RPMI complete medium in a ratio of 1:2 (T0 = pre-operative, T1 = upon arrival at intensive care unit, T2 = 6 h post-operatively, T3 = 24 h post-operatively) or 0.05 µg/mL LPS for 1 to 4 h. Stattic is well characterized and established in the scientific literature. Stattic specifically inhibits the JAK-STAT3 pathway by inhibiting the STAT3 SH2 domain and therefore STAT3 activation and nuclear translocation. It has been shown to be a reliable inhibitor of short-term cell stimulation with high specificity for inhibiting STAT3 without significantly affecting other cell signaling pathways, which allows for the investigation of the role of STAT3 [51].

### 4.3. Protein Isolation and Western Blot Analysis

For an intracellular protein analysis, cells were harvested after a 1 h stimulation and centrifuged at 6000× *g* for 10 min and cell pellets were lysed in an RIPA buffer supplemented with protease and phosphatase inhibitors (1:100, Sigma-Aldrich, Munich, Germany). Protein concentration was assessed via a Pierce BCA Protein Assay (Thermo Scientific, Karlsruhe, Germany). Intracellular protein samples were incubated with a Pierce Lane Marker Reducing Sample Buffer (Thermo Scientific, Karlsruhe, Germany) at 95 °C for 5 min and loaded onto a 12% SDS polyacrylamide gel for electrophoresis. Afterwards, proteins were transferred onto an Immobilon^®^ -P polyvinylidene fluoride membrane (Merck Millipore Ltd., Cork, Ireland) overnight at 30 V using a tank-blotting procedure (Bio-Rad Laboratories, Munich, Gernamy). Membranes were then blocked for 1 h at room temperature with 5% BSA (Pierce^TM^, Rockford, IL, USA) or 3% nonfat dried milk (PanReac AppliedChem, Darmstadt, Germany) in TBS + 0.1% Tween 20. Primary antibodies for β-Actin (1:20,000, Cell Signaling, Boston, MA, USA, Cat# 4967), phospho-NF-κB p65 (Ser536) (1:500, Cell Signaling, Boston, USA, Cat# 3033), NF-κB p65 (1:1000, Cell Signaling, Boston, USA, Cat# 3034), phosphor-Stat3 (Tyr705) (1:500, Cell Signaling, Boston, USA, Cat# 9145), and Stat3 (1:1000, Cell Signaling, Boston, USA, Cat# 9139) were diluted in a blocking solution and incubated overnight at 4 °C. Secondary antibodies anti-rabbit IgG-HRP (1:20,000, Cell Signaling, Boston, USA, Cat# 7074) or anti-mouse IgG-HRP (1:10,000, Santa Cruz, Heidelberg, Germany, Cat# sc-516102)) were incubated for 1 h at room temperature. SuperSignal™ West Dura Extended Duration Substrate (Thermo Fisher Scientific, Karlsruhe, Germany) was used to visualize protein expression, captured using a Molecular Imager^®^ ChemiDoc^TM^ XRS System, and Image Lab^TM^ Software V6.1.0 (Bio-Rad, Feldkirchen, Germany) was used for a densitometry analysis.

### 4.4. RNA Isolation and RT-PCR

Total RNA from THP-1 cells was isolated via acidic phenol/chloroform extraction using ROTI^®^Zol RNA (Carl Roth, Karlsruhe, Germany) according to the manufacturer’s instructions. RNA concentration and purity were assessed by spectrophotometric measurements at 260 nm and 280 nm with a Nanodrop 2000 (Nanodrop, Thermo Fisher Scientific, Karlsruhe, Germany) and agarose gel electrophoresis. Reverse transcription was performed using 500 ng total RNA via a qScriber^TM^ cDNA Synthesis Kit (highQu, Kraichtal, Germany) in a thermal cycler (PTC200, MJ Research, St. Bruno, Canada) according to the manufacturer’s instructions. The expression of target genes and GAPDH as a reference gene control was analyzed by real-time qPCR using TaqMan Gene Expression Assays (summarized in Table 2) and ORA^TM^ SEE qPCR Probe ROX H Mix (highQu, Kraichtal, Germany) according to manufacturers’ recommendations. We assessed the relative quantification of gene expression normalized to GAPDH as a reference gene via the 2^−∆∆Ct^ method and results are depicted as fold changes.

### 4.5. Statistical Analysis

A power analysis showed that a sample size of 51 participants will reach a power of 80% at a two-sided level of significance of 5% to detect a difference in TNFα with a Cohen’s d effect size of 0.4 (mean difference of −0.58 and standard deviation (SD) of 1.42 based on our preliminary data). The sample size calculation based on the paired *t*-test was used despite the intended analysis with a one-way repeated ANOVA. Regarding a post hoc test, it will lead to the sample size based on the paired *t*-test. Sample size calculation was performed by using program R version 3.6.2 with a package “pwr” and built-in command “pwr.t.test” (https://CRAN.R-project.org/package=pwr, accessed on 31 March 2023).

Western blot and RT-qPCR data were analyzed and illustrated using GraphPad Prism 10 (GraphPad Software, Inc., La Jolla, CA, USA). RM one-way ANOVA with Dunnett’s multiple comparisons post-test was relative to an individual pre-operative control (T0), a paired t-test was relative to an untreated sample of the same time point, or an unpaired t-test was relative to a control; * *p* ≤ 0.05 was considered significant. For the Western blot analysis, data from at least 3 independent experiments are presented as the mean ± SD and *p* values < 0.05 were considered significant. For a patient serum sample analysis, the relative quantification of gene expression normalized to GAPDH as a reference gene via the 2^-∆∆Ct^ method was conducted, and results are depicted as fold changes relative to T0. *p* values < 0.05 were considered significant.

## 5. Conclusions

In summary, we provide evidence that the inflammatory response following cardiac surgery involving CPB is driven in part by the activation of the STAT3 signaling pathway in macrophages. CPB induced an increased expression of cytokines and mediators of vascular permeability in macrophages via the activation of STAT3 by IL-6, IL-8, and IL-10 in the serum samples obtained post-operatively from CHD patients. Pretreatment with Stattic attenuates IL-1β, IL-10, and VEGF but promotes TNFα and SOCS3 expressions in macrophages. However, the effect of Stattic on ICAM expression was dependent on the duration of CPB. Our findings support the hypothesis that STAT3 is involved in the systemic inflammatory response observed in CHD patients following cardiac surgery involving CPB, resulting in both pro- and anti-inflammatory effects. Moreover, the patient population with congenital heart disease (CHD) includes individuals of a diverse age range and weight. We hypothesize that the factors contributing to post-operative inflammation, particularly the complexity of the CHD, surgical duration, and aortic clamping time, which is associated with ischemia–reperfusion-induced injury, have a greater impact on outcomes than age per se. Nonetheless, the limitations of our study to investigate the systemic inflammatory response induced by CPB lie in the heterogeneity of our cohort as well as the use of a monoculture in vitro model.

## Figures and Tables

**Figure 1 ijms-25-12398-f001:**
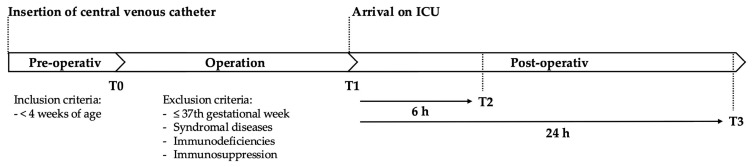
Patients’ blood samples were obtained via the central venous line pre-operatively after the induction of anesthesia (T0), post-operatively upon arrival in the pediatric intensive care unit (PICU) (T1), and both 6 h (T2) and 24 h (T3) after the operation.

**Figure 2 ijms-25-12398-f002:**
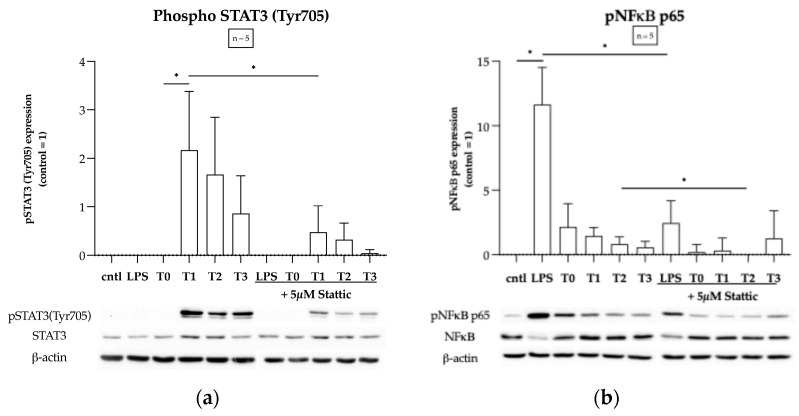
(**a**) CPB induces the activation of STAT3 at phosphorylation site Tyr705 that was attenuated by pretreatment with 5 µM Stattic in THP-1 macrophages in vitro. LPS had no effect on STAT3 phosphorylation. (**b**) CPB had no effect on the NF-κB p65 signaling pathway in THP-1 macrophages in vitro. Pretreatment with Stattic attenuated LPS-induced NF-κB p65 phosphorylation. Data from five separate experiments (*n* = 5) are represented as box plots. Statistical test analysis: RM one-way ANOVA with Dunnett’s multiple comparisons post-test relative to individual pre-operative control (T0), a paired t-test relative to an untreated sample of the same time point, or an unpaired *t*-test relative to control; * *p* ≤ 0.05 was considered significant.

**Figure 3 ijms-25-12398-f003:**
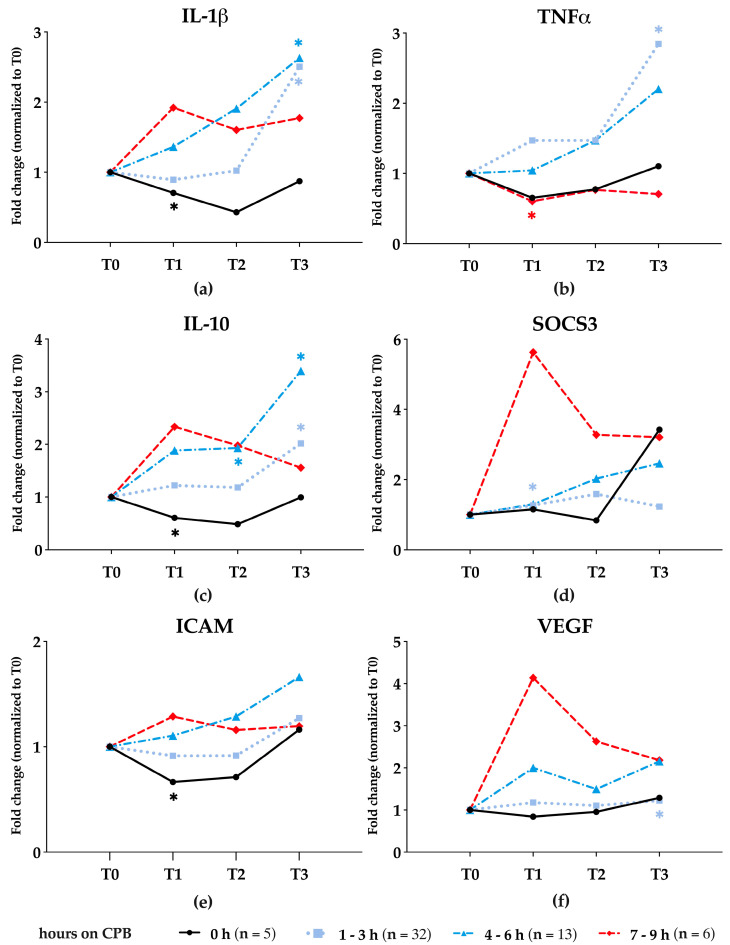
A prolonged duration of CPB correlates with expressions of serum-induced cytokines and mediators of vascular permeability in THP-1 macrophages in vitro, including (**a**) IL-1β, (**b**) TNFα, (**c**) IL-10, (**d**) SOCS3, (**e**) ICAM, and (**f**) VEGF. Data from 56 patients are represented as line graphs. Statistical test analysis: RM one-way ANOVA with Dunnett’s multiple comparisons post-test relative to patients not undergoing CPB (0 h); * *p* ≤ 0.05 was considered significant.

**Figure 4 ijms-25-12398-f004:**
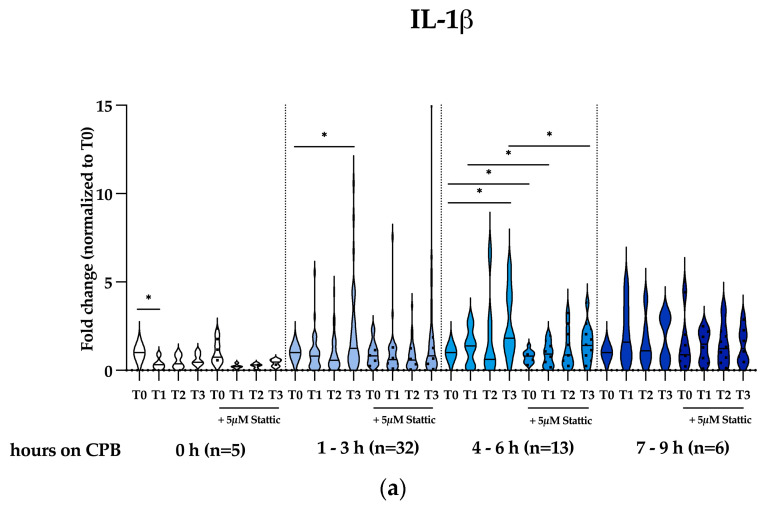
Stattic attenuates CPB-induced IL-1β and IL-10 expressions but augments TNFα expression in THP-1 macrophages in vitro. Serum samples from patients undergoing cardiac surgery without and with CPB for 1–3 h (*n* = 32) and 4–6 h (*n* = 13) induced significantly higher (**a**) IL-1β and (**c**) IL-10 expressions that could be attenuated by pretreatment with Stattic. No significant increases were observed in patients who did not require CPB (0 h, *n* = 5) nor in patients with an extremely long duration of CPB (7–9 h, *n* = 6). CPB for 1–3 h also elicited a significant increase in (**b**) TNFα expression that was augmented by pretreatment with Stattic. Data from 56 patients are represented as violin plots. Statistical test analysis: RM one-way ANOVA with Dunnett’s multiple comparisons post-test relative to individual pre-operative control (T0) or a paired *t*-test relative to an untreated sample of the same time point; * *p* ≤ 0.05 was considered significant.

**Figure 5 ijms-25-12398-f005:**
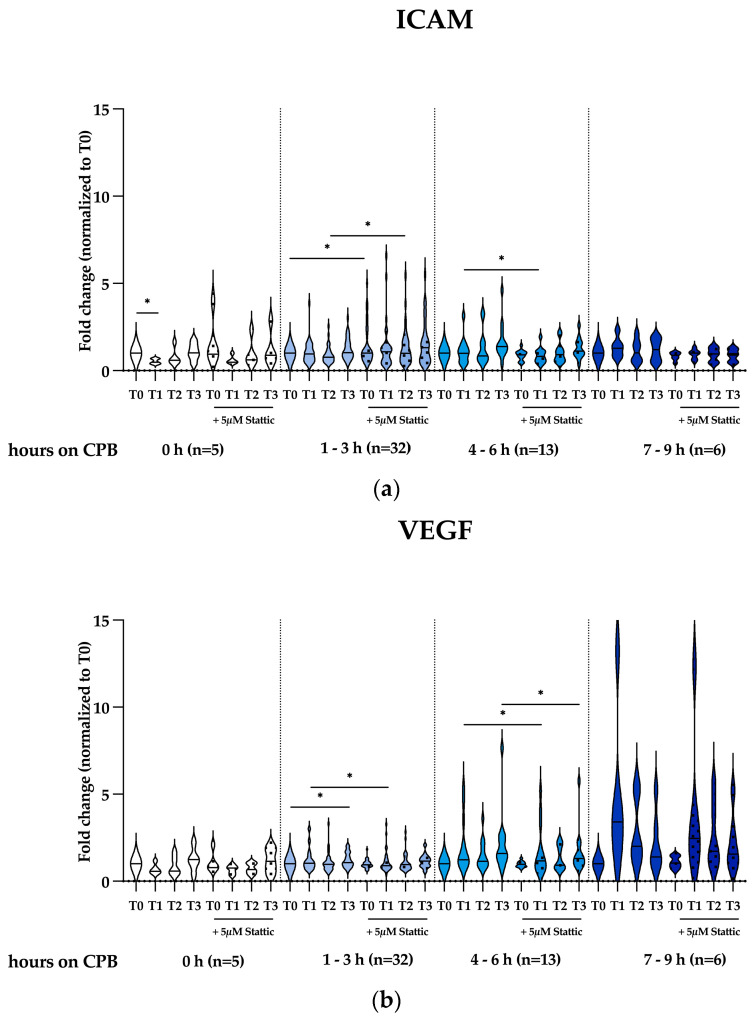
Effects of Stattic on expressions of mediators of vascular permeability, (**a**) ICAM, (**b**) VEGF, and (**c**) the inhibitor of STAT3 activation (SOCS3) in THP-1 macrophages in vitro, stimulated with serum samples from patients undergoing cardiac surgery without and with CPB. The patient cohort was classified into four distinct groups based on the duration of CPB as follows: 0 h (*n* = 5), 1–3 h (*n* = 32), 4–6 h (*n* = 13), and 7–9 h (*n* = 6). Statistical test analysis: RM one-way ANOVA with Dunnett’s multiple comparisons post-test relative to individual pre-operative control (T0) or a paired *t*-test relative to an untreated sample of the same time point; * *p* ≤ 0.05 was considered significant.

**Table 1 ijms-25-12398-t001:** Demographic and clinical data.

	Sex	
	Male	Female
Number of patients: 56	31 (55%)	25 (45%)
Demographic data (median; range)		
Age (years)	2.3 (0–59)	3.6 (0–65)
Weight (kg)	11.6 (2.8–101)	15.2 (3.2–104)
Characteristics of operation and CBP (median; range)		
RACHS-1	3 (1–4)	2 (1–6)
Warnes	2 (1–3)	2 (1–3)
Operation time (min)	361 (93–900)	313 (153–804)
CBP time (min)	180 (0–540)	120 (60–540)
Aortic cross-clamp time (min)	79 (0–518)	56 (0–266)

RACHS-1 = Risk Adjustment for Congenital Heart Surgery 1; CPB = cardiopulmonary bypass.

**Table 2 ijms-25-12398-t002:** List of TaqMan Real-Time PCR Assays for RT-qPCR analysis.

Gene	Assay ID
IL-1β	Hs01555410_m1
IL-10	Hs00961622_m1
TNFα	Hs00174128_m1
ICAM	Hs00164932_m1
VEGF	Hs00900055_m1
SOCS3	Hs02330328_s1
GAPDH	Hs02786624_g1

## Data Availability

The data presented in this study are available on request from the corresponding author. The data are not publicly available due to privacy or ethical restrictions.

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
