# Peer review of "Prolonged Extracorporeal Circulation Leads to Inflammation and Higher Expression of Mediators of Vascular Permeability Through Activation of STAT3 Signaling Pathway in Macrophages"

_ijms, 2024, doi:10.3390/ijms252212398_

Round 1

Reviewer 1 Report

Comments and Suggestions for Authors

As you described, children undergoing cardiac surgery with cardiopulmonary bypass (CPB) are at greater risk of postoperative complications due to the systemic inflammatory response caused by innate inflammatory mediators. However, the inclusion of males aged 0 to 50 years with an average age of 2.3, and females aged 0 to 54 years with an average age of 3.6, appears unusual (this applies to the weight range as well). Could you provide a clearer outline of the patient population?

Also, could you display fold change instead of delta delta Ct in Figures 3, 4, and 5?

Author Response

We thank the reviewer for the constructive comments and hope that we have satisfactorily addressed all of the comments from the reviewer.

Comment 1: As you described, children undergoing cardiac surgery with cardiopulmonary bypass (CPB) are at greater risk of postoperative complications due to the systemic inflammatory response caused by innate inflammatory mediators. However, the inclusion of males aged 0 to 50 years with an average age of 2.3, and females aged 0 to 54 years with an average age of 3.6, appears unusual (this applies to the weight range as well). Could you provide a clearer outline of the patient population?

Response 1: Thank you for pointing out this very relevant aspect of our cohort. We have included the following statements in the Introduction section (Lines 73-76 and 109-112, respectively) to better clarify our study cohort inclusion criteria and goal of the study. "Furthermore, the group of surviving adults with congenital heart defects (ACHD) is growing ever larger thanks to medical advances and treatments. ACHD represent a unique patient group also with distinct physiological characteristics that may impact both their surgical outcomes and inflammatory responses to CPB[11, 12]." and "The goal of our exploratory study is to better understand the heterogenous patient population who underwent corrective or palliative heart surgery for CHD at our hospital, as well as to shed light on the complex pathophysiology of CPB-induced inflammation in this cohort."   Moreover, we have also included the following statement in the Results section (Lines 127-131) to better describe our CHD patients cohort, "We included patients receiving their first and corrective operation such as closure of simple atrial or ventricular septal defects, as well as patients receiving palliative surgery such as Norwood I, pulmonal atrial banding or Fontan procedures in order to collect a representative cohort for patients with CHD."

Comment 2: Also, could you display fold change instead of delta delta Ct in Figures 3, 4, and 5?

Response 2: Agree. Thanks you for this suggestion. We have changed the labelling of our RT-qPCR graphs in Figures 3, 4, and 5, from  âˆ†âˆ†-Ct to "Fold change (normalised to T0)" as recommended. 

Reviewer 2 Report

Comments and Suggestions for Authors

The authors have designed a serum biomarker study in children undergoing cardiac surgery to better understand the inflammatory stress response.

 Aside from the paucity of explicit mention of ethics review, IRB, and consent process, the methodology is appropriate. The results support their conclusion.

 The reviewer applauds the authors for this study, which is appropriately designed and conducted. 

The manuscript is also well composed.

 1. Please place the methods section to follow the Introduction.  It would help the reader follow along the study if the methods section preceded the results.

 2. Please explicitly state whether the ethics review committee and/or IRB was involved.  Also state the consent process.  The reviewer recognizes that these were followed, based on the reference of a previous study #33.  Nevertheless, conducting human subject research, particularly on pediatric patients, is an important distinction that deserves the explicit mention of the diligent process each time.

This study is relatively novel in its design in that there are not many human subject studies in pediatric cardiac surgery patients, mostly due to the challenging nature of such a study design.  That by itself deserves special mention.

Author Response

We thank the reviewer for the kind review of our manuscript and for acknowledging the novelty and need of our study design. Please find the detailed responses below and the corresponding revisions/corrections highlighted/in track changes in the re-submitted files. We hope we have satisfactorily addressed all of the comments from the reviewer.

Comment 1. Please place the methods section to follow the Introduction.  It would help the reader follow along the study if the methods section preceded the results.

Response 1. Agree. Thank you for this comment. We also agree that it would be better if the Methods section preceded the Results. However, we used the manuscript Layout Template given by the journal for submission. Perhaps the layout will change in the last editing phase before publication. I will of course relay the request to the editor.

Comment 2. Please explicitly state whether the ethics review committee and/or IRB was involved.  Also state the consent process.  The reviewer recognizes that these were followed, based on the reference of a previous study #33.  Nevertheless, conducting human subject research, particularly on pediatric patients, is an important distinction that deserves the explicit mention of the diligent process each time.

Response 2. Agree. Thank you for pointing out this very important issue. We have included the following sentence at the beginning of the Methods section (Lines 412-413), "The study was approved by the Ethics Committee of Charité—Universitätsmedizin Berlin, Germany (decision EA2/180/19)."

Moreover, the following sections were included towards the end of the manuscript (Lines 544-554) as recommended by the journal's Layout Template:

Institutional Review Board Statement: The prospective observational feasibility study was conducted at the German Heart Center Berlin after approval by the Ethics Committee of Charité – Universitätsmedizin Berlin, Germany (decision EA2/180/19). The study was registered with the German register for clinical studies before patients recruitment (registration number: DRKS00020885; https://www.drks.de/drks_web/navigate.do?navigaionId=trial.HTML&TRIAL_ID=DRKS00020885).

Informed Consent Statement: Written consent was obtained from the parents of each patient before study inclusion. Patients younger than 18 years receiving a corrective or palliative cardiac surgery at our center were enrolled. Exclusion criteria were as follows: gestational age ≤ 37 weeks, a known maternal alcohol or substance abuse during pregnancy, immunodeficiencies or immunosuppressive medication, syndromic diseases (e.g. Trisomy 21 and 18), and congenital kidney disease.

Reviewer 3 Report

Comments and Suggestions for Authors

This paper demonstrates the role of the STAT3 signalling pathway in mediating postoperative inflammation during cardiopulmonary bypass (CPB) in paediatric patients. Enhanced recovery protocols and careful monitoring of cytokine levels are essential to manage inflammation. 

However, we have to question the methodology used to make this claim. Not only did they not provide references for their method, but they didn't explain it in detail, which makes it a bit difficult to understand. In other words, I would like to see the methodology described in more detail, such as how much serum was added to the THP-1 cells or how it was performed. 

It is unclear why stattic, a drug that inhibits the STAT3 protein, was used to culture THP-1 cells. There are many drugs that inhibit the STAT3 protein and it should be clearly explained why only this drug was used.

Table 1 and Lines 102-103: The study cohort consisted of 31 males of median age 2.3 years (range: 0 to 59 years) and 25 females of median age 3.6 years (range: 0 to 65 years). I don't understand the 2.3 years (range: 0 to 59 years) and 3.6 years (range: 0 to 65 years) part here. I don't understand the 2.3 years (range: 0-59) and 3.6 years (range: 0-65). 

Figure 3 is also quite difficult to understand because it is presented as a graph. It should be simplified to clearly show the differences.

The cytokine data is also somewhat difficult to understand: there are proinflammatory cytokines TNF-α, IL-6, and IL-8 and anti-inflammatory cytokines IL-10 and IL-1, and it is unclear if the data is limited to TNF-α, IL-1, and IL-10.

Regarding Figure 3, the proof using stattic seems to be the data in Figure 4 and Figure 5, but the data is not easy to understand. 

Overall, it seems that the authors tried to prove that all of these data are clinically meaningful by applying them to THP-1 cell culture, but it is questionable whether they presented the correct experimental model.

Comments on the Quality of English Language

Minor editing of English language required.

Author Response

We thank the reviewer for the thorough and constructive critique of our manuscript. Please find the detailed responses below and the corresponding revisions/corrections highlighted/in track changes in the re-submitted files.We hope we have satisfactorily addressed all of the comments from the reviewer.

Comment 1. This paper demonstrates the role of the STAT3 signalling pathway in mediating postoperative inflammation during cardiopulmonary bypass (CPB) in paediatric patients. Enhanced recovery protocols and careful monitoring of cytokine levels are essential to manage inflammation. 

However, we have to question the methodology used to make this claim. Not only did they not provide references for their method, but they didn't explain it in detail, which makes it a bit difficult to understand. In other words, I would like to see the methodology described in more detail, such as how much serum was added to the THP-1 cells or how it was performed. 

Response 1. Thank you for pointing out this important aspect of our methodology. We have included the following changes to our Methods: Cell culture protocol (Lines 422-438), 

"THP-1 cells represent a human monocyte cell line that can be transformed into macrophages using PMA, as described below. By establishing a transformation and activation protocol in our laboratory, the use of THP-1 cells allows us to obtain reproducible results and to mimic human inflammatory responses under controlled conditions [49, 50]. Immortalized human THP-1 cells were purchased from ATCC (American Type Culture Collection: TIB-202) and grown in suspension in RPMI 1640 (Gibco) medium supplemented with 10 % (v/v) heat inactivated fetal bovine serum (FBS, Biochrom) and 100 U/ml penicillin and 100 µg/ml streptomycin (Gibco) in a humidified 37 °C, 5 % CO2 incubator. 1 x 106 cells were plated in 35 x 10 mm tissue culture treated plates (NunclonTM Delta Surface, Thermo Scientific) in the presence of 15 ng/mL phorbol 12-myristiate-12 acetate (PMA, Sigma-Aldrich) for 3 days, followed incubation in PMA free medium for 24 h prior to stimulation with serum samples or LPS.  Differentiated THP-1 macrophages were then pretreated with 5 µM Stattic, a STAT3 (Tyr705) inhibitor, for 1 h, followed by incubation with serum samples, which were diluted with RPMI complete medium in a ratio of 1:2 (T0 = preoperative, T1 = upon arrival at intensive care unit, T2 = 6 h, T3 = 24 h postoperative) or 0.05 µg/mL LPS for 1 to 4 h."  

Comment 2. It is unclear why stattic, a drug that inhibits the STAT3 protein, was used to culture THP-1 cells. There are many drugs that inhibit the STAT3 protein and it should be clearly explained why only this drug was used.  

Response 2. Thank you for this comment. We have included the following statement with literature citation in the Methods: Cell culture protocol (Lines 438-443) to justify our use of Stattic,  

"Stattic is well characterized and established in the scientific literature. Stattic specifically inhibits the JAK-STAT3 pathway by inhibiting the STAT3 SH2 domain and therefore, STAT3 activation and nuclear translocation. It has been shown to be a reliable inhibitor of short-term cell stimulation with high specificity for inhibiting STAT3 without significantly affecting other cell signaling pathways, which allows for investigation of the role of STAT3 [52]."  

Comment 3. Table 1 and Lines 102-103: The study cohort consisted of 31 males of median age 2.3 years (range: 0 to 59 years) and 25 females of median age 3.6 years (range: 0 to 65 years). I don't understand the 2.3 years (range: 0 to 59 years) and 3.6 years (range: 0 to 65 years) part here. I don't understand the 2.3 years (range: 0-59) and 3.6 years (range: 0-65).   

Response 3. Thank you for pointing out this very relevant aspect of our cohort. We wanted to investigate potential CPB-induced inflammatory response in a general pupulation of CHD patients including both children and adults. So we have included the following statements in the Introduction section (Lines 73-76 and 109-112, respectively) to better clarify our study cohort inclusion criteria and goal of the study.

["Furthermore, the group of surviving adults with congenital heart defects (ACHD) is growing ever larger thanks to medical advances and treatments. ACHD represent a unique patient group also with distinct physiological characteristics that may impact both their surgical outcomes and inflammatory responses to CPB[11, 12]." and "The goal of our exploratory study is to better understand the heterogenous patient population who underwent corrective or palliative heart surgery for CHD at our hospital, as well as to shed light on the complex pathophysiology of CPB-induced inflammation in this cohort."]  

Moreover, we have also included the following statement in the Results section (Lines 127-131) to better describe our CHD patients cohort,

["We included patients receiving their first and corrective operation such as closure of simple atrial or ventricular septal defects, as well as patients receiving palliative surgery such as Norwood I, pulmonal atrial banding or Fontan procedures in order to collect a representative cohort for patients with CHD." ]

Comment 4. Figure 3 is also quite difficult to understand because it is presented as a graph. It should be simplified to clearly show the differences.

Response 4. Thank you for this valuable critique of our data on cytokines and mediators of vascular permeability expressions in THP-1 macrophages illustrated in Figure 3. The purpose of this Figure is to show the kinetics of expression of the targets investigated throughout the preoperative time points relative to duration of CPB to convey our findings that, "Prolonged duration of CPB correlates with serum-induced cytokines and mediators of vascular permeability expressions in THP-1 macrophages in vitro." This is summarized in the Result section (Lines 270-275),

["In summary, increasing duration of CPB from 1-6 h elicit an immune response that gradually increased post-operatively upon reaching significancy relative to individual baseline values 24 h after the operation. In general, the 4-6 h CPB group showed a higher immune response than the shorter 1-3 h group. On the other hand, extreme duration of CPB lasting from 7-9 h resulted in an immediate induced immune response in the macrophages upon arrival at the ICU (T1) that gradually decreased towards baseline."]

Comment 5. The cytokine data is also somewhat difficult to understand: there are proinflammatory cytokines TNF-α, IL-6, and IL-8 and anti-inflammatory cytokines IL-10 and IL-1, and it is unclear if the data is limited to TNF-α, IL-1, and IL-10.

Response 5. Thank you for this comment and we apologise that the graphs were difficult to understand. Figure 4 illustrates cytokines expressions (IL-1b, TNFa and IL-10) and Figure 5 illustrates expressions of the mediators of vascular permeability (ICAM and VEGF), well as the inhibitor of STAT3 activation (SOCS3) induced by the patients serum samples organised according to duration of CPB in the THP-1 macrophages. Additionally, the figures also illustrates the effect of blocking the STAT3 signaling pathway by Stattic. In summary, we observed (Figure 4 and Figure 5, respectively):

"Figure 4. Stattic attenuates CPB-induced IL-1b and IL-10 expression but augments TNFa expression in THP-1 macrophages in vitro. Serum samples from patients undergoing cardiac surgery without and with CPB for 1-3 h (n=32) and 4-6 h (n=13) induced significantly higher (a) IL-1b and (c) IL-10 expression that could be attenuated by pre-treatment with Stattic. No significant increases were observed in patients who did not require CPB (0h, n=5) nor in patients with extremely long duration of CPB (7-9 h, n=6). CPB for 1-3 h also elicit significant increase in (b) TNFa expression that was augmented by pre-treatment with Stattic. Data from 56 patients are represented as violin plots. Statistical test analysis: RM one-way ANOVA with Dunnett’s multiple comparisons post-test relative to individual pre-operative control (T0) or paired t-test relative to untreated sample of the same time point, *p ≤ 0.05 was considered significant."

"Figure 5. Effects of Stattic on mediators of vascular permeability (a) ICAM, (b) VEGF and (c) the inhibitor of STAT3 activation (SOCS3) expressions in THP-1 macrophages in vitro stimulated with serum samples from patients undergoing cardiac surgery without and with CPB. The patient cohort was classified into four distinct groups based on duration of CPB as followed: 0h (n=5), 1-3 h (n=32), 4-6 h (n=13) and 7-9 h (n=6). Statistical test analysis: RM one-way ANOVA with Dunnett’s multiple comparisons post-test relative to individual pre-operative control (T0) or paired t-test relative to untreated sample of the same time point, *p ≤ 0.05 was considered significant."

We hope this satisfactorily clarify our findings illustrated in the figures. 

Comment 6. Regarding Figure 3, the proof using stattic seems to be the data in Figure 4 and Figure 5, but the data is not easy to understand. 

Response 6. Thank you for this valid comment. Once again we apologize for any confusion or misunderstanding of the data illustrated in the figures. Figure 3 is intended to show the kinetics of the inflammatory response induced by duration of CPB throughout the preoperative time without the effect of Stattic. Figures 4 and 5 show in more details the effect of CPB on the inflammatory response, including distribution and mean of the patients cohort, as well as the accent of Stattic on the response for the targets investigated.  Indeed, there is a lot of data from our experimental findings that we wanted to report and it was not easy to decide on which type of graphs/style that would accomplish this in the most efficient and easy to understand way. In the end, we believe the figures submitted in the manuscript to describe our very complex data set best illustrates our findings in the most comprehensible way.  We hope this will be satisfactory for the reviewer.

Comment 7. Overall, it seems that the authors tried to prove that all of these data are clinically meaningful by applying them to THP-1 cell culture, but it is questionable whether they presented the correct experimental model.

Response 7. Thank you for this valid comment. We agree that using the THP-1 cell culture model is not ideal, compared to perhaps using the collected serum samples to stimulate ex vivo each individual patient's PBMC harvested before the operation. However, such cell model is not possible as we are not able to withdraw the large amount of blood needed from the majority of our small patients in our cohort. Moreover, using a primary macrophage model would introduce batch purification variability to our study due to the large amount of serum samples we needed to test. Therefore, as stated in our cell culture protocol, we feel that the THP-1 cells were a reasonable model for this study.

"THP-1 cells represent a human monocytic cell line that can be transformed to macrophages by stimulating with PMA, as further described below. By optimizing a standard transformation and activation protocol in our laboratory, the use of THP-1 cells allows us to obtain reproducible results that mimic human inflammatory responses under controlled conditions [50, 51]."

Additionally, we have revised the purpose of our exploratory study to not mislead the readers into thinking our findings can be directly translated to the clinic, as stated in the Introduction (Lines 109-112),

"The goal of our exploratory study is to better understand the heterogenous patient population who underwent corrective or palliative heart surgery for CHD at our hospital, as well as to shed light on the complex pathophysiology of CPB-induced inflammation in this cohort."

Comment 8. Minor editing of English language required.

Response 8. Thank you for this suggestion. We have revised the manuscript for English spelling and grammatical errors. We hope we have satisfactorily addressed all of the reviewer's comments.

Round 2

Reviewer 1 Report

Comments and Suggestions for Authors
  • For Figure 2, in addition to showing pSTAT3, it would be important to include total STAT3 in the blot, rather than just actin as a loading control."

  • I noticed that you changed the y-axis in Figure 3 from ΔΔCt to fold change, but the results themselves were not adjusted accordingly.  Could you please clarify this aspect?"

Comments on the Quality of English Language

Pretty good

Author Response

Comment 1. For Figure 2, in addition to showing pSTAT3, it would be important to include total STAT3 in the blot, rather than just actin as a loading control."

Response 1. Agree. Thank you for this suggestion. We have changed the graphs accordingly and hope evrerything is now satisfactory.

Comment 2. I noticed that you changed the y-axis in Figure 3 from ΔΔCt to fold change, but the results themselves were not adjusted accordingly.  Could you please clarify this aspect?

Response 2. Agree. Thank you for pointing out this discrepancy in the graphs of our qPCR data. The original submitted graphs were indeed illustrations of the RQ values but were mistakenly labeled as ΔΔCt instead of 2-ΔΔCt, which also can be interpreted as fold change. Therefore, the resubmitted graphs show the same RQ or Fold change data with the y-axis, this time properly labeled. This was indeed a major blunder on our part and we sincerely apologize for this mistake. We will be happy to submit our qPCR raw data if there are any further questions regarding the data. 

Reviewer 3 Report

Comments and Suggestions for Authors

I believe it is publishable with the appropriate revision.

Author Response

Comment 1. I believe it is publishable with the appropriate revision.

Response 1. Thank you for taking the time to thoroughly review our manuscript. Your comments/recommendations have greatly enhanced our manuscript.